# Research on Current Backflow of Asymmetric CHB Multi-Level Inverter

**Manyuan Ye ***[ID]**, Guizhi Song, Wei Ren and Qiwen Wei**

College of Electrical and Automation Engineering, East China Jiaotong University, Nanchang 330013, China; SGZ_318@163.com (G.S.); renwei_1995@163.com (W.R.); weiqiwens@163.com (Q.W.)
* Correspondence: myye@ecjtu.edu.cn

**Abstract:** For the traditional asymmetric cascaded H-bridge multi-level inverters, the conventional hybrid modulation method has the problem of current backflow in a certain modulation index range. Although the monopolar modulation method effectively solves this problem, the high-voltage unit participates in the high-frequency operation in part of the range, which limits the improvement of the switching frequency of the whole system. The hybrid frequency modulation method can reduce the switching frequency of the high voltage unit to a certain extent, but the harmonic characteristics of the output voltage will be affected. In order to solve the above problems, a double frequency modulation method based on level-shifted PWM (LS-PWM) is proposed. On the one hand, it solves the inherent power back filling problem of the traditional hybrid modulation method, on the other hand, it ensures that the output voltage of the inverter has good harmonic characteristics when the switching frequency of the high voltage unit is low. The results of simulation and experiment prove the correctness of the theoretical analysis.

**Keywords:** cascaded H-bridge multilevel inverters; frequency modulation; logic operation; harmonic characteristics; current backflow

## 1. Introduction

Cascaded H-bridge (CHB) multi-level inverters are one of the most popular inverters in the field of high-voltage and high-power converters and AC drives, which has the advantages of output multiple voltage levels, excellent harmonic performance, and easy modular design and manufacture [1–3]. With the development of multilevel technology, hybrid cascaded multilevel inverters have been widely used. Its main circuit topology is also a cascade structure of H-bridge units, but the voltage level and switching frequency of each H-bridge unit are different [4–6]. For instance, a new asymmetric cascaded multilevel inverter [7] (ACHB) proposed by Manjreka et al. has two H-bridges with an input DC voltage ratio of 2:1. The high-voltage H-bridge consists of IGCT and the low-voltage H-bridge consists of IGBT. The structure can output more levels in the same cascade unit, so it can reduce the number of power switch devices and DC power supply in the same voltage level, at the same time, it can allow a variety of different voltage levels of power switches to work together, which increases the flexibility of control. Its design idea is in line with the development characteristics of power electronic devices, and can be well-applied in electric vehicles, photovoltaic [8–10], and other applications.

Aiming at this type of asymmetric cascaded multi-level inverters, the following modulation strategies are proposed: phase-shifted Pulse Width Modulation (PS-PWM) is proposed in reference [11], which has the advantage of balancing the power of each unit; the level-shifted PWM (LS-PWM) modulation strategy proposed in reference [12] can obtain high quality sinusoidal waveforms; reference [13] proposes to use the traditional hybrid modulation (H-PWM) strategy to make high-voltage unit H1 work at the fundamental frequency, and low-voltage unit H2 work at the high frequency,

which can enormously reduce the switching losses. However, in the above three modulation strategies, there exists a moment when the polarity of the output voltage of the two units is opposite, the current backflow problem will occur in the low-voltage unit. For this reason, reference [14] uses space vector modulation to modulate hybrid cascaded 7-level inverters, and eliminates energy circulation by choosing appropriate switching states, but the modulation process is too complex. Reference [15] adopts the specific harmonic elimination (SHEPWM) control, but the complex calculation of switching angle limits the application of this method. Reference [16] proposes an improved hybrid frequency modulation strategy, which solves the problem of current inversion, but the low switching frequency of the high voltage unit affects the quality of the output voltage of the inverter to a certain extent. In reference [17], by improving the traditional carrier amplitude shift modulation method and taking advantage of the fact that the output voltage phases of the two units are always the same, the problems of current backflow existing in the traditional hybrid modulation method are effectively solved.

Based on the above research, a double frequency modulation strategy based on LS-PWM is proposed for the traditional type II ACHB inverter topology. This strategy not only can solve the problems of current backflow and energy backflow inherent in the traditional hybrid modulation, but also improve the problem of relatively poor harmonic characteristics caused by the high voltage unit working in the low frequency under the hybrid frequency modulation, so as to reduce the content of low harmonic and improve the output power quality of the inverter.

This paper is organized as follows: the hybrid seven-level topology and its operating principle are presented in Section 2. Section 3 introduces the modulation principle in references [13,16,17]. The double frequency modulation strategy based on LS-PWM is proposed to solve the problem of power backflow in Section 4. Simulation and experimental results are given in Sections 5 and 6 to validate the performance of the proposed method. Finally, the conclusions are summarized in Section 7.

## 2. Topology of Hybrid Seven-Level Inverter

Figure 1 shows the topological structure of Type II ACHB inverter. The DC voltage ratio of the two cascaded units is 2:1, and the output phase voltage $u_{AN}$ is:

$$u_{AN} = u_{H1} + u_{H2} \tag{1}$$

where $u_{H1}$ and $u_{H2}$ are the output voltage of the high-voltage unit H1 and the low-voltage unit H2 respectively, and $i_0$ is the output current of the inverter.

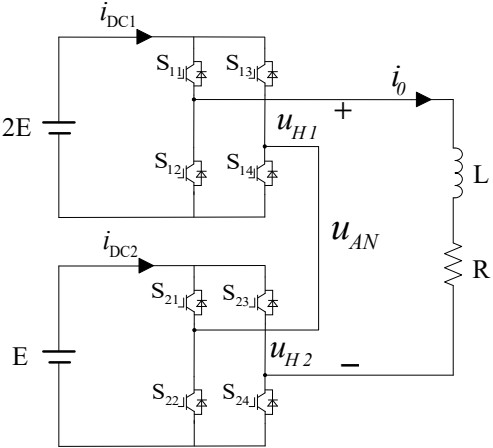

**Figure 1.** Topology of hybrid seven-level inverter.

According to the operating principle of the inverter, each cascade unit can output three different levels of voltage. For high-voltage unit H1, $u_{H1}$ can output +2E, −2E, and 0, and for low voltage-unit H2, $u_{H2}$ can output +E, −E, and 0. Therefore, the output phase voltage of the inverters can reach 7

levels, which are (+3E), (+2E), (+E), and 0, respectively. Suppose that the switching function of unit i (i = 1, 2) is:

$$S_i = \begin{cases} 1, & u_{Hi} = Ei \\ 0, & u_{Hi} = 0 \\ -1, & u_{Hi} = -Ei \end{cases} \tag{2}$$

Then the output level of each unit can be expressed as:

$$u_{Hi} = S_i E \tag{3}$$

where, $S_i$ denotes the switching function of the first unit, and $u_{Hi}$ denotes the output level of the second unit, and $i$ = 1, 2.

Therefore, the total output level of cascaded inverters is:

$$u_{AN} = \sum_{i=1}^{2} u_{Hi} \tag{4}$$

The state of cascaded inverter and output voltage levels of each unit is represented by (S1, S2). The output voltage levels of each unit are shown in Equation (3) and the total output voltage levels of cascaded inverters are shown in Equation (4). As it can be seen from the synthesis method of PWM levels in each interval in Figure 2 that when different PWM synthesis methods are selected, the output waveforms of different PWM levels will be corresponding, so that the output voltage waveforms of all levels of joint units are different. In the following, the positive half cycle is taken as an example to elaborate the synthesis method of PWM level in each interval.

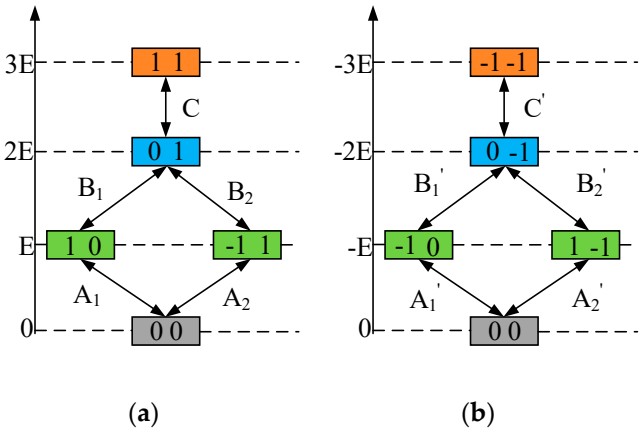

(**a**)                    (**b**)

**Figure 2.** Synthesis of PWM level in different intervals. (**a**) Positive half period. (**b**) Negative half period.

(1) Interval [0, E]: When the output voltage is at the PWM level of interval [0, E], there are two ways to synthesize it. In the mode of A1 [(0, 0) ~ (+1, 0)]: alternating output 0 and E, constant output 0; in the mode of A2: alternating output 0 and - E, alternating output 0 and 2E. For working mode A1, low-voltage unit H2 works in high frequency PWM state, high-voltage unit H1 works in low frequency PWM state, and the polarity of output voltage of the two units is the same. For mode A2 [(0,0) ~ (−1,+1)], low-voltage unit H2 and high-voltage unit H1 all work in high frequency state, and their output voltage polarity is opposite.

(2) Interval [E, 2E]: There are also two types of synthesis methods in this region. In Mode B1[(+1,0) ~ (0,+1)]: High frequency PWM levels of alternating output E and 0, and high frequency PWM levels of alternating output 0 and 2E, both of which have the same polarity; in Mode B2 (−1, +1) ~ (0, +1): High frequency PWM levels of alternating output −E and 0, and low frequency PWM levels of constant output 2E, the polarity of which is opposite.

(3) Interval [2E, 3E]: In this region, there is only one synthesis mode C[(0,+1) ~ (+1,+1)]: alternating output 0 and E, working in high frequency PWM state, while $u_{H1}$ constant output 2E, working in low frequency PWM state, the polarity of output voltage of the two units is the same.

## 3. Hybrid CHB Optimized Modulation Strategy

Because the paper mainly deals with the current inversion problem of type II ACHB inverters, the application of space vector modulation and SHE-PWM technology is limited. Therefore, the present paper mainly describes the hybrid frequency modulation strategy proposed in reference [16] and the unipolar LS-PWM modulation strategy proposed in reference [17], as well as the double frequency modulation strategy based on LS-PWM proposed in the present paper.

### 3.1. Traditional Hybrid Modulation Strategy

The principle of traditional hybrid modulation strategy proposed in reference [13] is shown in Figure 3.

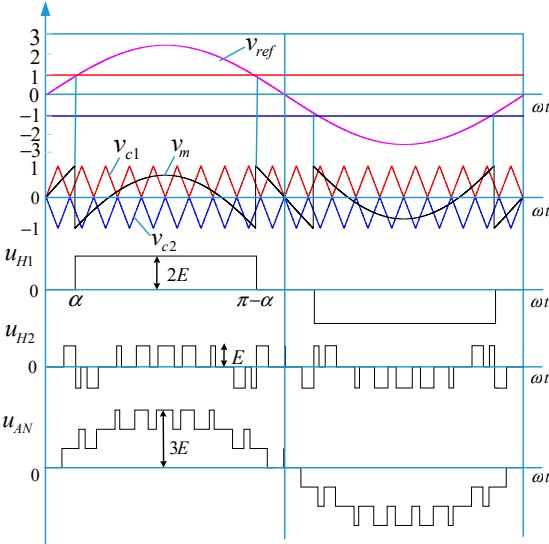

**Figure 3.** Schematic diagram of traditional mixed modulation strategy. (Reference [13]).

In the traditional hybrid modulation strategy proposed in reference [13], the fundamental wave amplitude of the output phase voltage $u_{AN}$, the output voltage $u_{H1}$ of H1 unit, and the output voltage $u_{H2}$ of H2 unit can be obtained as follows:

$$u_{AN(1)} = 3EM \sin \omega t \tag{5}$$

$$u_{H1(1)} = \begin{cases} 0, & 0 \le M \le 1/3 \\ \frac{8E}{\pi} \sqrt{1 - \frac{1}{9M^2}} \sin \omega t, & 1/3 \le M \le 1 \end{cases} \tag{6}$$

$$u_{H2(1)} = \begin{cases} 3EM \sin \omega t, & 0 \le M \le 1/3 \\ \left(3EM - \frac{8E}{\pi} \sqrt{1 - \frac{1}{9M^2}}\right) \sin \omega t, & 1/3 \le M \le 1 \end{cases} \tag{7}$$

where, M is the modulation index, $u_{H1(1)}$ and $u_{H2(1)}$ are the fundamental amplitude of high-voltage unit H1 and low-voltage unit H2 output voltage respectively, and $u_{AN(1)}$ is the fundamental amplitude of the output phase voltage of the inverter. The relationship between the fundamental amplitude of three voltages and the modulation M is shown in Figure 4. It can be seen that when M is in the interval [0, 1/3], the output voltage of H1 is 0, and the output power of the inverter is all borne by H2. When M is located in the interval [0.37, 0.78], the polarity of the output voltage of the two units is

opposite. High-voltage unit H1 injects energy into low-voltage unit H2, resulting in the problem of current back-filling and energy back-flow. At this moment, $u_{h1}$ is larger than $u_{AN}$, $u_{H2}$ is less than 0, therefore the output power of H1 unit exceeds the requirement of load and affects the stability of DC bus capacitor voltage.

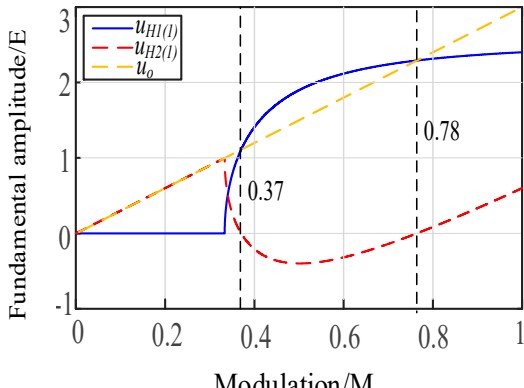

**Figure 4.** The relationship between the amplitude of the fundamental wave and the modulation index M.

### 3.2. Hybrid Frequency Modulation Strategy

The schematic diagram of the hybrid frequency modulation strategy is shown in Figure 5. The method adopts positive and negative superposition modulation, and makes the carrier frequency of the low-voltage unit H2 twice that of the high-voltage unit H1. Assuming that the carrier frequencies of H2 and H1 are $f_{crl}$ and $f_{crh}$ respectively, then $f_{crl} = 2f_{crh}$. The switching pulse S$_{11}$ and S$_{13}$ of H1 unit are obtained by comparing the modulated wave $v_m$ with the carrier $v_{crh}$ of the positive and negative phases of the middle two layers, and then the switching pulse S$_{11}$ and S$_{13}$ of H2 unit are obtained by comparing with the other four layers of carrier $v_{crl}$. It can be concluded that under HF-PWM strategy, high-voltage unit H1 works in low frequency PWM state, low-voltage unit H2 works in high frequency PWM state, and the output voltage of the two units has the same polarity in positive and negative half-cycle, thus effectively solving the problems of current backfilling and energy feedback. However, because of the reduction of H1 switching frequency, the harmonic performance of the output voltage of the inverter is relatively poor, which affects the improvement of power quality.

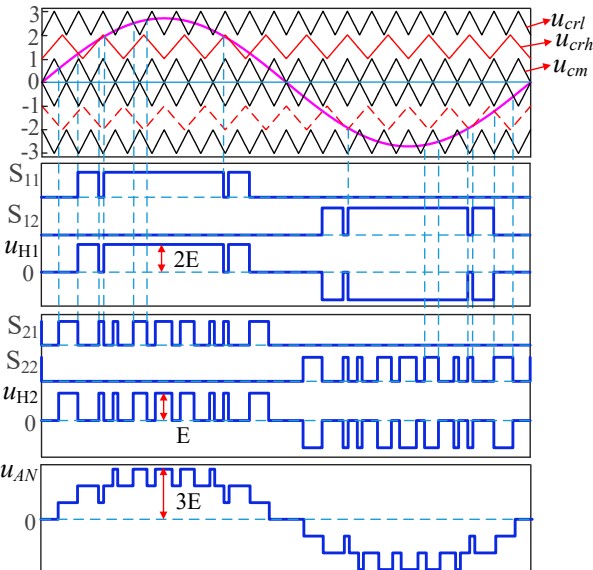

**Figure 5.** Hybrid frequency modulation schematic (Reference [16]).

### 3.3. Unipolar LS-PWM Modulation Strategy

The modulation diagram of the unipolar LS-PWM method proposed in literature [17] is shown in Figure 6. The reference sinusoidal signal $u_{cm}$ is compared with three triangular carriers $u_{cr1}$, $u_{cr2}$, and $u_{cr3}$, which are located above the zero reference line and arranged continuously from top to bottom, to obtain three logical pulse signals A, B, C. At the same time, the reference sinusoidal signal $-u_{cm}$, which is opposite to $u_{cm}$, is compared with triangular carriers $u_{cr1}$, $u_{cr2}$, and $u_{cr3}$, to get A', B', C'. The polar signal R of the modulated wave is obtained by comparing the signal $u_{cm}$ directly with the zero reference voltage. R is always high in the positive half period of the modulated wave and zero in the negative half period. Because of the unipolar modulation characteristic of this method, the driving signals of left-arm switches $S_{11}$, $S_{12}$, $S_{21}$, and $S_{22}$ are determined by polar signals R, while the driving signals of the other switches are determined by R and six other logical pulse signals.

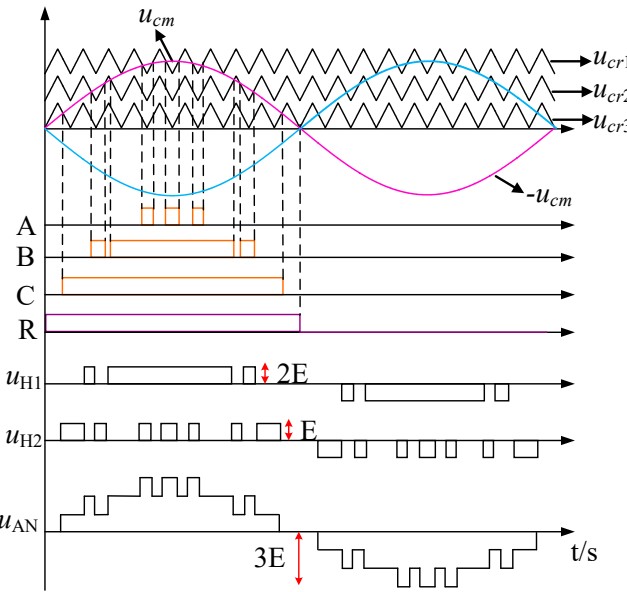

**Figure 6.** Unipolar level-shifted PWM (LS-PWM) modulation strategy schematic diagram (Reference [17]).

The relationship between the fundamental amplitude of three voltages and the modulation M is shown in Figure 7. It can be seen that the polarity of the output fundamental wave voltage of the high voltage unit and the output fundamental wave voltage of the low voltage unit are always positive in the whole modulation ratio range. Therefore, when utilizing unipolar LS-PWM modulation, there is no power back-filling problem in the whole modulation ratio range.

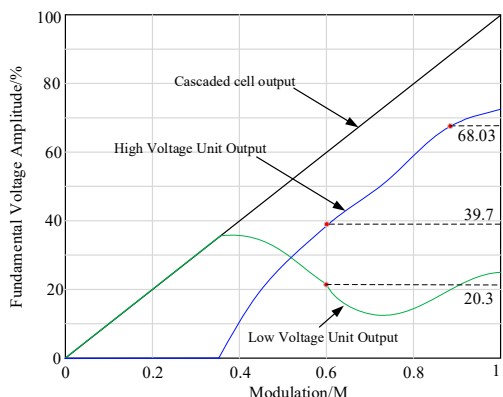

**Figure 7.** Unipolar LS-PWM modulation strategy schematic diagram (Reference [17]).

## 4. Double Frequency Modulation Strategy Based on LS-PWM

In view of the problems of the above-mentioned hybrid frequency modulation strategy, the principle of double frequency modulation strategy based on LS-PWM proposed in this paper is shown in Figure 8. First, the sinusoidal wave absolutely deserves the modulation wave $v_m$. Then, the pulse signals $A_1$, $A_2$, and $A_3$ are obtained by comparing with the three-layer carrier $v_{cr1}$, $v_{cr2}$, and $v_{cr3}$, and the pulse signal Q is obtained by comparing with the constant value E. The polar signal R is generated by the zero-crossing point of $v_m$, which means the polarity signal R is 1 in the first half cycle and 0 in the second half cycle. The amplitude of carrier $v_{cr1}$ and $v_{cr2}$ is 2E, the frequency is 1/2 of that of $v_{cr3}$, and the phase difference is 180 degrees, so H1 can output high frequency PWM waveform at lower switching frequency. Finally, the pulse driving signals $S_{11}$, $S_{14}$, $S_{21}$, and $S_{24}$ of H1 and H2 switches are obtained by combining the obtained pulse signals $A_1$, $A_2$, and $A_3$ logically. Assuming that the frequency of carrier $v_{cr1}$ and $v_{cr2}$ is $f_{ch}$ and the frequency of carrier $v_{cr3}$ is $f_{cl}$, then $f_{cl} = 2 f_{ch}$. The generation of $S_{11}$, $S_{14}$, $S_{21}$, and $S_{24}$ pulse signals of switches is described in detail below:

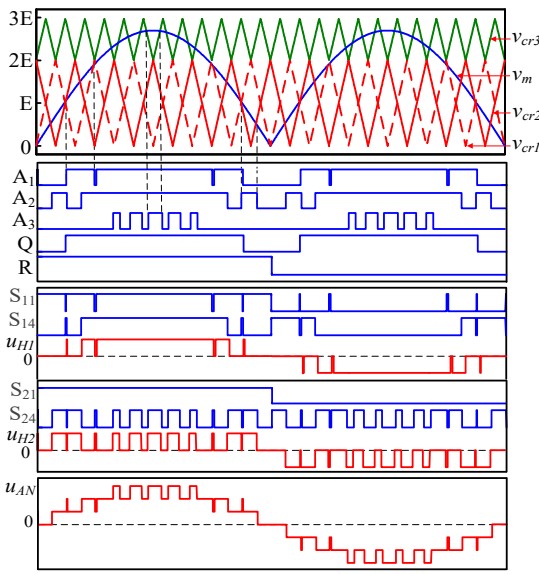

**Figure 8.** Schematic diagram of double frequency modulation based on LS-PWM.

1) In the positive half cycle: For high-voltage unit H1, in region [0, E], switch $S_{11}$ is constantly turned on and $S_{14}$ is constantly turned off. In region [E, 2E], when $E < v_m < v_{cr1}$, $S_{11}$ output is 1, and when $E < v_m < v_{cr2}$, $S_{11}$ output is 0. When $E < v_m < v_{cr2}$, $S_{14}$ output is 0, and when $v_{cr2} < v_m < 2E$, $S_{14}$ output is 1. Within the region [2E, 3E], both $S_{11}$ and $S_{14}$ outputs are 1; for low-voltage unit H2, the output of switch $S_{21}$ is constant to 1 throughout the region. In region [0, E], when $v_{cr1} < v_m < E$ or $v_{cr2} < v_m < E$, the output of $S_{24}$ is 1, and vice versa is 0. In the region [2E, 3E], when $v_m > v_{cr3}$, the output of $S_{24}$ is 1, and vice versa is 0. Combined with the above analysis, the pulse signals of positive half-cycle switches are obtained as follows:

$$\begin{cases} S_{11} = \overline{Q} + A_1 \\ S_{12} = \overline{\overline{Q} + A_1} \\ S_{13} = \overline{QA_2} \\ S_{14} = QA_2 \end{cases} \begin{cases} S_{21} = R \\ S_{22} = \overline{R} \\ S_{23} = \overline{\overline{Q}(A_1 + A_2) + Q(\overline{A_1} + \overline{A_2}) + A_3} \\ S_{24} = \overline{Q}(A_1 + A_2) + Q(\overline{A_1} + \overline{A_2}) + A_3 \end{cases} \qquad (8)$$

2) In the negative half-cycle: Combining with the analysis of 1), the pulse signals of each switch in negative half-cycle can be obtained as follows:

$$\begin{cases} S_{11} = \overline{\overline{Q} + A_1} \\ S_{12} = \overline{Q} + A_1 \\ S_{13} = QA_2 \\ S_{14} = \overline{QA_2} \end{cases} \begin{cases} S_{21} = R \\ S_{22} = \overline{R} \\ S_{23} = \overline{Q}(A_1 + A_2) + Q(\overline{A_1} + \overline{A_2}) + A_3 \\ S_{24} = \overline{\overline{Q}(A_1 + A_2) + Q(\overline{A_1} + \overline{A_2}) + A_3} \end{cases} \tag{9}$$

Combining 1) and 2) mentioned above, and using polarity signal R to summarize the pulse signals of positive and negative half cycle switches, the pulse signals of each switch in the whole cycle can be obtained as follows:

$$\begin{cases} S_{11} = R(\overline{Q} + A_1) + \overline{R}(\overline{\overline{Q} + A_1}) \\ S_{12} = R(\overline{Q} + A_1) + \overline{R}(\overline{\overline{Q} + A_1}) \\ S_{13} = RQA_2 + \overline{R}(\overline{QA_2}) \\ S_{14} = RQA_2 + \overline{R}(\overline{QA_2}) \end{cases} \tag{10}$$

$$\begin{cases} S_{21} = R \\ S_{22} = \overline{R} \\ S_{23} = \dfrac{R[\overline{Q}(A_1 + A_2) + Q(\overline{A_1} + \overline{A_2}) + A_3]+}{\overline{R}[\overline{\overline{Q}(A_1 + A_2) + Q(\overline{A_1} + \overline{A_2}) + A_3}]} \\ S_{24} = R[\overline{Q}(A_1 + A_2) + Q(\overline{A_1} + \overline{A_2}) + A_3]+ \\ \overline{R}[\overline{\overline{Q}(A_1 + A_2) + Q(\overline{A_1} + \overline{A_2}) + A_3}] \end{cases} \tag{11}$$

As shown in Figure 8, the expression of modulation wave $v_m$ is as follows:

$$v_m = 3EM\big|\sin(\omega t)\big| \tag{12}$$

where, M is the modulation index.

Therefore, the Fourier series of modulated wave $v_m$ can be expressed as:

$$F(\omega t) = \frac{3EM}{\omega}\left(\frac{2}{\pi} - \frac{4}{\pi}\sum_{n=1}^{\infty}\frac{\cos(n\omega t)}{4n^2 - 1}\right) \tag{13}$$

Assuming that the carrier frequency is much higher than the modulation frequency, the relationship between the output voltage of the high-voltage unit and its modulation wave can be obtained by the state space averaging method:

$$u_{H1} = \frac{2E}{3EM}F(\omega t) \tag{14}$$

where $u_{H1}$ is the output voltage of high-voltage unit H1.

Similarly, the output fundamental voltage $u_{AN(1)}$ of hybrid cascaded H-bridge seven level inverter is:

$$u_{AN(1)} = 3EM \sin \omega t \tag{15}$$

Then the output voltage $u_{H2}$ of the low-voltage unit H2 is:

$$u_{H2} = u_{AN} - u_{H1} \tag{16}$$

$$u_{H2(1)} = u_{AN(1)} - u_{H1(1)} \tag{17}$$

where, $u_{H1(1)}$ and $u_{H2(1)}$ are the fundamental amplitude of high-voltage unit H1 and low-voltage unit H2 output voltage respectively, and $u_{AN(1)}$ is the fundamental amplitude of the output phase voltage of the inverter.

The relationship between the fundamental amplitude of output voltage and phase voltage of each unit and modulation index M under the frequency doubling modulation strategy based on LS-PWM proposed is shown in Figure 9.

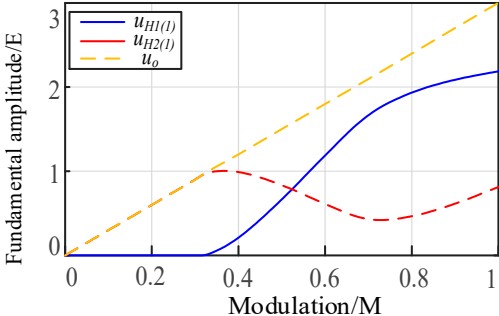

**Figure 9.** The relationship between the amplitude of the fundamental wave and the modulation index M.

Compared with Figure 4, it can be concluded that in the range of full modulation index, the fundamental amplitude of low-voltage unit H2 output voltage is all positive, and the fundamental amplitude of high-voltage unit H1 output voltage is not greater than that of phase voltage, which shows that the proposed strategy effectively solves the problems of current backfilling and energy reflux.

Table 1 shows the comparison among the four modulation strategies in terms of carrier number, output voltage THD value, and whether the current backflow can be solved.

**Table 1.** Comparison of different modulation method.

| Modulation Method | Number of Carriers | Modulated Wave ($v_m$) | Output of Low Voltage Unit ($u_{H2}$) | THD of Output Phase Voltage ($u_{AN}$) | Whether There is Current Backfilling |
|---|---|---|---|---|---|
| Reference 13 | 2 | 3Msin($wt$) | negative when M∈ [0.37, 0.78] | 19.97% | existence |
| Reference 16 | 6 | 3Msin($wt$) | positive when M∈ [0, 1] | 22.49% | Non-existent |
| Reference 17 | 3 | \|3Msin($wt$)\| | positive when M∈ [0, 1] | 22.37% | Non-existent |
| Proposed in this paper | 3 | \|3Msin($wt$)\| | positive when M∈ [0, 1] | 22.19% | Non-existent |

## 5. Simulation Result

In order to verify the effectiveness of double frequency modulation strategy based on LS-PWM, the hybrid frequency modulation strategy, the unipolar LS-PWM modulation strategy proposed in reference [16] and the optimized frequency doubling modulation strategy are modeled and simulated in Matlab/Simulink respectively. The key parameters of the model are shown in Table 2.

**Table 2.** Key parameters of the Simulink model.

| Parameter | Value |
|---|---|
| DC power (E/V) | 50 |
| Resistance (R/Ω) | 24 |
| Inductance (L/H) | 0.004 |
| Carrier frequency ($f_{crl}, f_{cl}$/kHz) | 4 |
| Output frequency ($f$/Hz) | 50 |
| Modulation (M) | 0.3, 0.6, 0.9 |
| CTNR (N) | 80 |

When the modulation index are 0.3, 0.6, and 0.9, respectively, the output characteristics of hybrid cascaded H-bridge inverters are obtained under hybrid frequency modulation strategy, unipolar LS-PWM modulation strategy proposed in reference [16], and double frequency modulation strategy based on LS-PWM.

The output characteristics of hybrid cascaded H-bridge inverters with modulation index of 0.3, 0.6, and 0.9 are shown in Figure 10a–c, respectively, under hybrid frequency modulation strategy, unipolar LS-PWM modulation strategy and double frequency modulation strategy based on LS-PWM.

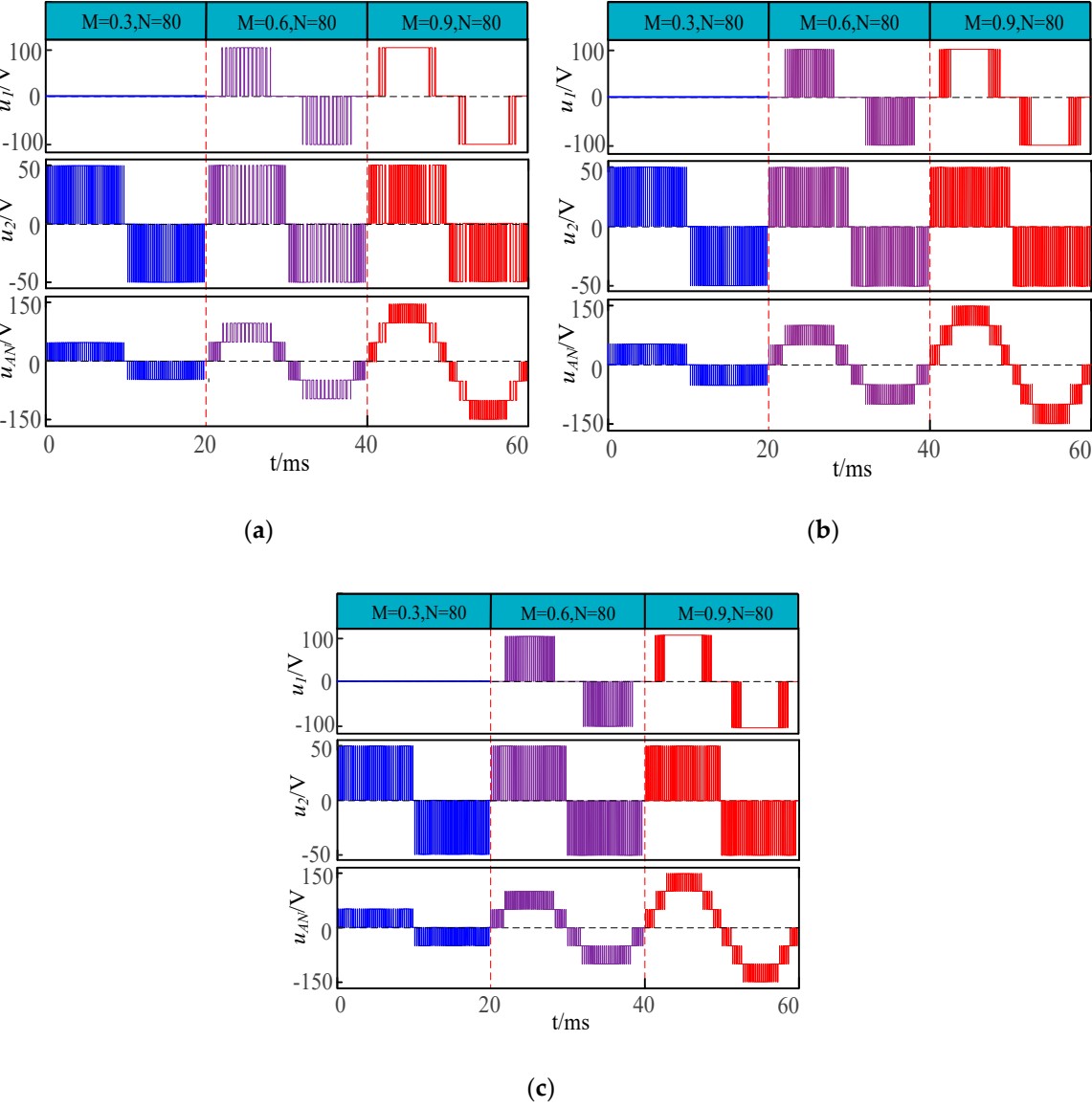

**Figure 10.** Voltage output charts under three modulation strategies. (**a**) Output characteristics under mixed frequency modulation strategy. (**b**) Output characteristics under unipolar LS-PWM modulation strategy. (**c**) Output characteristics of double frequency modulation strategy based on LS-PWM.

The output voltage and phase voltage waveforms of each cell under the mixed frequency modulation strategy are shown in Figure 10a. It can be seen from the figure that in the positive half-cycle the output voltage and phase voltage of high-voltage unit H1 and low-voltage unit H2 are all positive polarities, in the negative half-cycle the output voltage and phase voltage of high-voltage unit H1 and low-voltage unit H2 are all negative polarities, and the output polarities of the two units are the same throughout the cycle, thus effectively solving the problems of current backfilling and energy

feedback caused by traditional hybrid modulation. However, because of the low carrier frequency of high-voltage unit H1, the harmonic characteristics of its output voltage will also be affected, which increases the low-order harmonic content of the inverter output phase voltage and reduces the power quality of the phase voltage.

The output voltage and phase voltage waveforms of each unit under the "unipolar LS-PWM modulation strategy" mentioned in reference [16] are shown in the Figure 10b. Its effect is basically the same as the double frequency modulation strategy based on LS-PWM shown in Figure 10c, both of them can solve the problems of current backfilling and energy reflux, but its high-voltage unit is high frequency modulation that will increase the switching loss.

The output voltage and phase voltage waveforms of each unit based on LS-PWM frequency doubling modulation strategy are shown in Figure 10c. It can be seen from the figure that the polarity of the output voltage of the two units is the same in the positive and negative half-cycle, which effectively solves the inherent problems of current backfilling and energy feedback of the traditional hybrid modulation, and $u_{AN}$ is closer to the standard seven-level. Moreover, the equivalent switching frequency of H1 and H2 is doubled to realize the double frequency control of phase voltage waveform at lower switching frequency. Therefore, the low-order harmonic performance of the inverter output phase voltage caused by the reduction of H1 switching frequency under mixed frequency modulation is effectively solved. At the same time, the harmonic characteristics of the inverter output voltage are improved, and the power quality of the inverter output is improved.

The spectrum distribution of the output phase voltage $u_{AN}$ of the inverter under three strategies with modulation index of 0.9 and carrier ratio of 60 is shown in Figure 11. Under the hybrid frequency modulation strategy, the fundamental wave amplitude of $u_{AN}$ is 134.8V and the THD is 22.49%. Under the unipolar LS-PWM modulation strategy proposed in reference [16], the fundamental wave amplitude of $u_{AN}$ is 134.9V and the THD is 22.37%. Under the double frequency modulation strategy based on LS-PWM, the fundamental wave amplitude of $u_{AN}$ is 135V, and the THD is 22.19%. It can be seen that, compared with the hybrid modulation strategy and the unipolar LS-PWM modulation strategy, the fundamental amplitude of the output voltage of the inverter increases relatively, the THD decreases relatively, the harmonics are uniformly distributed near the integer multiple of the carrier ratio, and the content of the low-order harmonics is greatly reduced, thus simplifying the design of the filter. Therefore, the double frequency modulation strategy based on LS-PWM improves the harmonic performance of the output voltage of the inverter as well as the power quality.

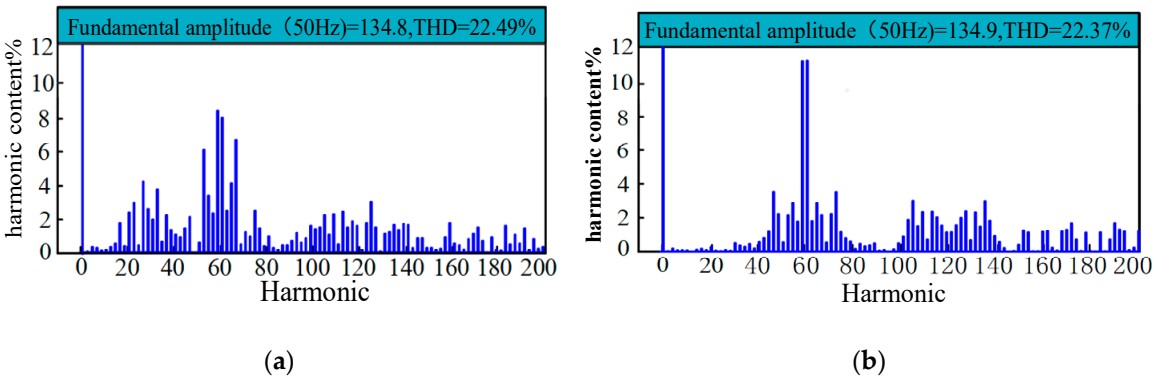

(**a**)          (**b**)

**Figure 11.** *Cont.*

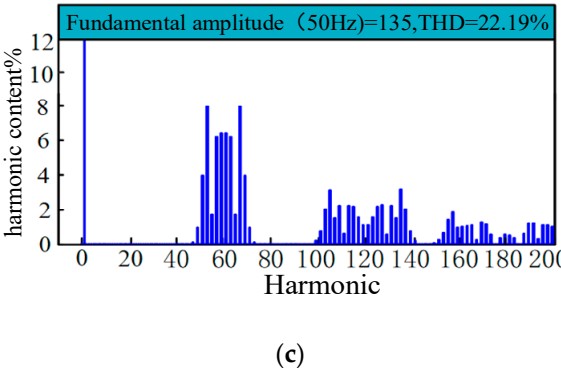

**(c)**

**Figure 11.** Spectrum of phase voltages under three strategies. (**a**) Phase voltage spectrum of hybrid frequency strategy. (**b**) Phase voltage spectrum of unipolar LS-PWM strategy. (**c**) Phase voltage spectrum of unipolar LS-PWM double frequency strategy.

## 6. Experiment Result

In order to verify the effect of the proposed control method, a hybrid seven-level experimental platform is established, which is controlled by DSP, and the experimental setup is presented in Figure 12. Experiments are carried out at the same carrier frequency with different modulation indexes and different carrier frequencies with the same modulation indexes. The main parameters of the system are shown in Table 3.

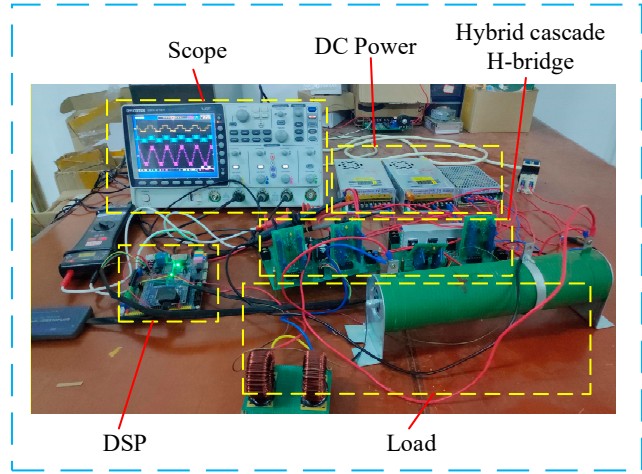

**Figure 12.** Experiment platform of hybrid cascaded 7-level inverter.

**Table 3.** Key parameters of the experiment.

| Parameter | Value |
| --- | --- |
| DC power (E/V) | 50,100 |
| Resistance (R/$\Omega$) | 20 |
| Inductance (L/H) | 0.002 |
| Carrier frequency ($f_{crl}$, $f_{cl}$/kHz) | 3.5 |
| Output frequency ($f$/Hz) | 50 |
| Modulation (M) | 0.6, 0.9 |
| CTNR (N) | 60 |

Figure 13a,b are the output voltage of each unit, the output voltage waveform of each unit and the spectrum of the output phase voltage when the modulation index is 0.9 and the carrier frequency is 5 KHz, respectively.

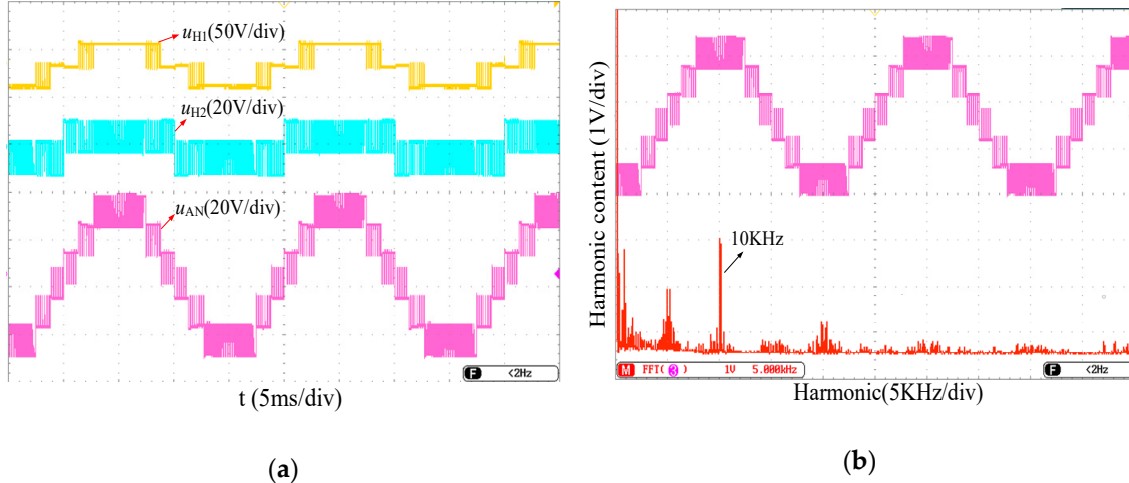

(**a**)

(**b**)

**Figure 13.** Experiment result (m = 0.9, frl = 5 KHz). (**a**) Output voltage and phase voltage waveform of each unit. (**b**) Phase voltage spectrum.

Figure 13a shows that there is no moment of opposite voltage polarity in the output voltage waveforms of $u_{H1}$ and $u_{H2}$ of the two cells, which demonstrates that the double frequency modulation strategy based on LS-PWM can effectively solve the current inversion problem of type II ACHB inverters, and when the modulation index is 0.9, the output phase voltage is 7 levels. From Figure 13b, it can be seen that the harmonics of the output phase voltage are mainly concentrated near the carrier frequency, so high quality waveforms can be obtained under this modulation.

Figure 14a,b are the output voltage of each unit, the output voltage waveform of each unit and the frequency spectrum of the output phase voltage $u_{AN}$ when the modulation index is 0.9 and the carrier frequency is 3 KHz, respectively.

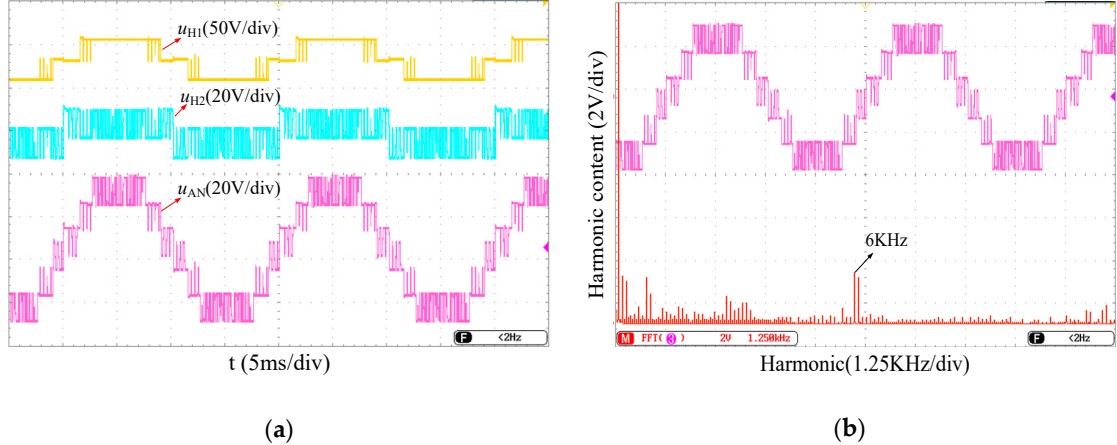

(**a**)

(**b**)

**Figure 14.** Experiment result (m = 0.9, frl = 3 KHz). (**a**) Output voltage of each unit in two cycles. (**b**) Phase voltage spectrum.

From Figure 14a, it can be concluded that the frequency doubling modulation strategy based on LS-PWM can effectively solve the current inversion problem of type II ACHB inverters. When the carrier frequency is changed to 3 KHz, the THD value of the output voltage is larger than that of the carrier frequency of 5 KHz, but the main harmonics are also near the carrier frequency.

Figure 15a,b shows respectively the output voltage of each unit, the output voltage waveform of each unit and the spectrum of the output phase voltage $u_{AN}$ when the modulation index is 0.6 and the carrier frequency is 3 KHz.

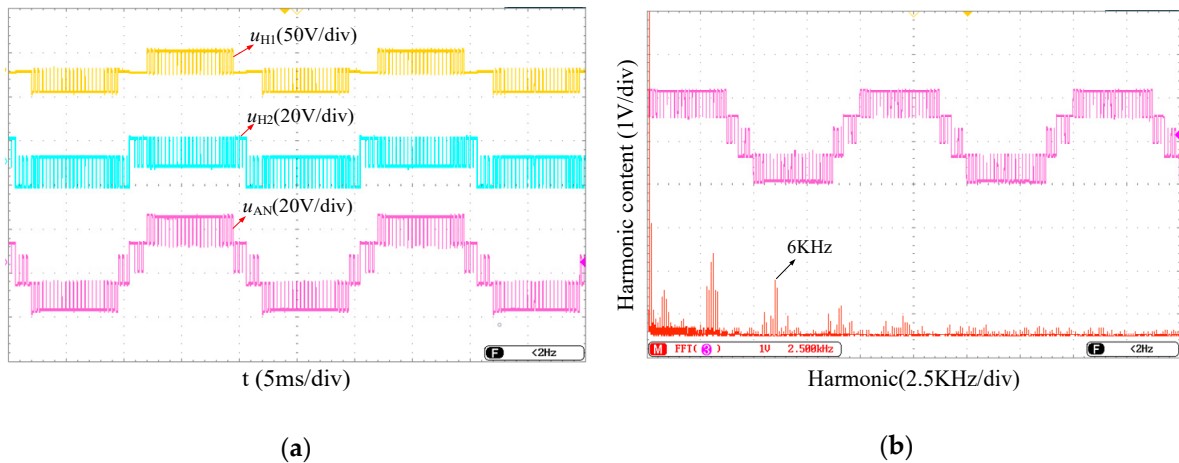

**Figure 15.** Experiment result (m = 0.6, frl = 3 KHz). (**a**) Output voltage of each unit in two cycles. (**b**) Phase voltage spectrum.

It can be seen from Figure 15a,b that the frequency doubling modulation strategy based on LS-PWM can still effectively solve the current inversion problem of type II ACHB inverters at low modulation index, but the output phase voltage is only 5 levels. It can be seen from Figure 15b that the harmonic of the output phase voltage $u_{AN}$ is mainly near the carrier, so high quality waveform can be obtained under this modulation method.

From Figures 13–15, it can be seen that the proposed frequency doubling modulation strategy based on LS-PWM can effectively solve the current inversion problem of type II ACHB inverters at any modulation index. At the same time, it can reduce the low frequency harmonic content on the basis of guaranteeing the quality of the output phase voltage waveform of the inverters, and the output voltage waveform quality is better when the carrier frequency is higher.

## 7. Conclusion

The paper shows that the hybrid frequency modulation strategy of the traditional type II ACHB inverter are improved, are verified by experiments, and the following conclusions are obtained:

(1) This strategy can effectively solve the problems of current backflow and energy feedback inherent in the traditional hybrid modulation under the full modulation index.

(2) Compared with the mixed frequency modulation strategy, the proposed double frequency modulation strategy based on LS-PWM can eliminate the low harmonics and get higher quality waveforms. Compared with the single polarity LS-PWM strategy, the double frequency modulation strategy based on LS-PWM reduces the switching frequency and the switching loss of the high voltage unit.

**Author Contributions:** Project management, M.Y.; Conceptualization, G.S.; methodology, G.S. and M.Y.; Analysision, G.S.; Software writing, W.R.; validation, G.S., W.R., and Q.W.; writing—original draft preparation, G.S.; writing—review and editing, M.Y. All authors have read and agreed to the published version of the manuscript.

**Funding:** This research was funded by National Natural Science Foundation of China, grant number 51767007; Jiangxi Provincial Industrial Science and Technology Support Project, grant number 20192BBEL50011; Jiangxi Natural Science Foundation Project, grant number 20192BAB206036; Jiangxi Provincial Youth Science Foundation of China, grant number GJJ180306.

**Conflicts of Interest:** There are no conflicts of interest regarding the publication of this paper.

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
