# Peer review of "Research on Current Backflow of Asymmetric CHB Multi-Level Inverter"

_electronics, doi:10.3390/electronics9020214_

Round 1

Reviewer 1 Report

Manyuan Ye and co-authors investigated a frequency double modulation based on LS-PWN modulation to solve the current backflow problem, reduce the content of low harmonic issue which exist in current hybrid frequency modulation. The authors first introduced the traditional hybrid modulation strategy and its major drawbacks. Then, they illustrated the advantages of double modulation based on LS-PWN modulation, the corresponding simulation results from these three methods were presented respectively. It has been demonstrated that the proposed method could effectively solve inherent problems of current backfilling and energy feedback and greatly reduce the content of low-order harmonics. To further verify this method, experiment was carried out with different modulation index and carrier frequency. The experimental results proved that the proposed method is able to solve the current inversion problem and reduce the low frequency harmonic content in the meantime. The method is illustrated thoroughly with both simulation and experimental results. Therefore, I agree with the publication of this manuscript.

Author Response

Thank you very much for your review, and I have made a comprehensive revision and embellishment of the paper.

Reviewer 2 Report

-I suggest editing the title of the paper. A research? what type of research?

-Please clearly state what is the novelty of this paper.

-The following papers are related to your paper and referring them will strengthen your paper:

a) Asymmetrical Cascaded Multi-Level Inverter Using Control Freedom Pulse Width Modulation Techniques

b) Feedforward Accurate Power Sharing and Voltage Control for Multi-Terminal HVDC Grids

-Please provide a table to present your experimental setup parameters.

-The conclusion needs to be improved.

Author Response

Review 2:

(1). Page 1, Line 2-3

I suggest editing the title of the paper. A research? what type of research?

I'm sorry that I didn't think about it comprehensively when I edited the title. In this paper, I mainly study the problem of current backflow in asymmetric cascaded H-bridge inverter. To solve this problem, I propose a double frequency modulation method based on LS-PWM. The reedited title has been modified in the document.

(2). Page 1, Line 9-20ï¼›Page 14, Line 356-362

Please clearly state what is the novelty of this paper.

There are some problems in the methods of solving the problem of current backflow. For example, there are many low order harmonics in the output voltage waveform of hybrid modulation, which are difficult to eliminate in practical application. The single polarity ls-pwm method improves the switching frequency of high-voltage unit devices and increases the switching loss. Based on these problems, this paper proposes a frequency doubled modulation method based on ls-pwm. The switching pulse of the high-voltage unit is obtained by using two triangular carriers with a phase difference of 180 ° with a lower frequency through PWM modulation and through logical combination, and finally realizes the effect of frequency doubling modulation with a smaller switching frequency, which not only solves the problem of current back-flow, but also avoids the existence of many low-order harmonics in the output voltage waveform. Compared with unipolar LS-PWM, this method reduces switching losses.

(3). Page 14, Line 371-373

The following papers are related to your paper and referring them will strengthen your paper:

Thank you very much for the papers recommended by the experts, which have benefited me a lot. I have added these papers to the references and hope that more people will learn from them.

(4). Page 12, Line 312-313

Please provide a table to present your experimental setup parameters.

Thank you very much for your valuable suggestions. I have added a table to describe the parameters of the experimental setup.

(5). Page 14, Line 356-362

The conclusion needs to be improved.

Thank you very much for your valuable suggestions. I have revised the conclusion of this paper and focused on the innovation of this paper.

Reviewer 3 Report

This manuscript presents a frequency doubling modulation method based on LS-PWM for an asymmetric CHB multi-level inverter. To support the work there is a comparison with the other methods. Moreover, besides the explanation of the proposed methods, the other ones are also explained. The work is also supported with simulation and experimental tests. From the analysis of the work appeared several question as following described:

- In page 7 is said “Then, the pulse signals A1, A2 and A3 are obtained by comparing with the three-layer carrier vcr1, vcr2 and vcr3,”. I suppose that what is compared with the with the three-layer carrier is the modulation wave and not the pulse signals

- In pag 7 line 196 there is a reference to switch tube S11. What is switch tube ?

- Regarding the polar signal R I suppose that is a binary signal that is 1 when the modulation wave is positive and 0 when that wave is negative. This should be clarified since a polarity signal an also be 1 and -1

- Comparing the proposed strategy with the one of reference 17, it is possible to see that they are practically the same. There is only a very minor difference regarding the output voltage THD. So, which can be considered the advantages and/or improvments of the proposed strategy ? Another aspect is related with the efficiency. There is any difference regarding this between those two modulation strategies ?

- The English requires a carefully revision

Author Response

Review 3:

(1). Page 7, Line 185-186

In page 7 is said “Then, the pulse signals A1, A2 and A3 are obtained by comparing with the three-layer carrier vcr1, vcr2 and vcr3,”. I suppose that what is compared with the with the three-layer carrier is the modulation wave and not the pulse signals.

I'm sorry that the sentence didn't describe clearly. The three signals A1, A2 and A3 are obtained by comparing the three carriers vcr1, vcr2 and vcr3 with the modulation wave vm respectively.

(2). Page 7, Line 196

In pag 7 line 196 there is a reference to switch tube S11. What is switch tube?

This is an error that I didn't check when I translated it. It has been revised in the paper at present.

(3). Page 7, Line 188-189

Regarding the polar signal R I suppose that is a binary signal that is 1 when the modulation wave is positive and 0 when that wave is negative. This should be clarified since a polarity signal an also be 1 and -1.

The polarity signal R here is 1 in the first half of the modulation wave vm and 0 in the second half of the modulation wave vm.

(4). Page 7, Line 182-189

Comparing the proposed strategy with the one of reference 17, it is possible to see that they are practically the same. There is only a very minor difference regarding the output voltage THD. So, which can be considered the advantages and/or improvments of the proposed strategy? Another aspect is related with the efficiency. There is any difference regarding this between those two modulation strategies?

The problem in reference 17 is that the frequency of the triangular carrier which generates the switching pulse signal of the high voltage unit and the low voltage unit is the same, that is to say, the high voltage unit is also the high frequency modulation, which improves the switching frequency of the high voltage unit and increases the switching loss of the high voltage unit. In this paper, the double frequency modulation method based on LS-PWM is proposed. The triangular carrier used to generate the switching pulse signal of high voltage unit is half of the carrier frequency of low voltage unit, so the switching frequency of high voltage unit can be reduced. For example, the carrier frequency of document 17 and the carrier frequency of double frequency modulation method based on LS-PWM proposed in this paper are both 10kHz, but the carrier frequency of high voltage unit in double frequency modulation method based on LS-PWM proposed in this paper is only 5KHz, so the proposed double frequency modulation method based on LS-PWM is more efficient. And when the two methods have the same carrier frequency (the carrier frequency of the double frequency modulation method based on LS-PWM proposed in this paper refers to the carrier frequency of the low-voltage unit), the output THD value of the method proposed in this paper is smaller.

Round 2

Reviewer 3 Report

This revision version received clarifications and improvements. Besides that, all the questions have been addressed. Thus, I do not have further questions.